# Research on a Cooperative Adaptive Cruise Control (CACC) Algorithm Based on Frenet Frame with Lateral and Longitudinal Directions

**DOI:** 10.3390/s23041888

**Published:** 2023-02-08

**Authors:** Pingli Ren, Haobin Jiang, Xian Xu

**Affiliations:** 1School of Automotive and Traffic Engineering, Jiangsu University, Zhenjiang 212013, China; 2School of Vehicle Engineering, Changzhou Vocational Institute of Mechatronic Technology, Changzhou 213164, China; 3Chery New Energy Automobile Co., Ltd., Wuhu 241000, China

**Keywords:** intelligent connected vehicle group, cooperative adaptive cruise control (CACC), the Frenet frame, decoupling control, trajectory tracking

## Abstract

Research on the cooperative adaptive cruise control (CACC) algorithm is primarily concerned with the longitudinal control of straight scenes. In contrast, the lateral control involved in certain traffic scenes such as lane changing or turning has rarely been studied. In this paper, we propose an adaptive cooperative cruise control (CACC) algorithm that is based on the Frenet frame. The algorithm decouples vehicle motion from complex motion in two dimensions to simple motion in one dimension, which can simplify the controller design and improve solution efficiency. First, the vehicle dynamics model is established based on the Frenet frame. Through a projection transformation of the vehicles in the platoon, the movement state of the vehicles is decomposed into the longitudinal direction along the reference trajectory and the lateral direction away from the reference trajectory. The second is the design of the longitudinal control law and the lateral control law. In the longitudinal control, vehicles are guaranteed to track the front vehicle and leader by satisfying the exponential convergence condition, and the tracking weight is balanced by a sigmoid function. Laterally, the nonlinear group dynamics equation is converted to a standard chain equation, and the Lyapunov method is used in the design of the control algorithm to ensure that the vehicles in the platoon follow the reference trajectory. The proposed control algorithm is finally verified through simulation, and validation results prove the effectiveness of the proposed algorithm.

## 1. Introduction

Currently, with the growth of automobile ownership, automobile accidents, environmental pollution, traffic congestion, and other problems have become increasingly serious [1]. Although advanced driver assistance systems including AEB, LKA, and LDW have the potential to reduce the occurrence of traffic accidents, it is difficult to deal with traffic congestion and pollution in the environment. As the first choice to solve such problems, the intelligent transport system (ITS) has gradually become the dominant direction in the field of transportation [2,3]. Guerrero-lbanez J et al. investigated how to integrate sensor technology with transportation infrastructure in order to achieve a sustainable Intelligent Transportation System (ITS) [4]. Cafiso S et al. proposed that the dynamic use of hard shoulder running (HSR) would represent a valid solution that both exploits existing infrastructures and facilitates traffic outflow by the implementation of smart digital roads, with limited interventions on the physical infrastructure [5]. With the development of intelligent technology and networking technology, it has become a new trend to integrate intelligent connected vehicles (ICV) into ITS. Based on sensors, controllers, actuators, and other devices, ICVs have the functions of environmental awareness, intelligent decision-making, and collaborative control to achieve safe, comfortable, and energy-saving autonomous driving. ICVs can improve traffic safety and road capacity and reduce energy consumption through information exchange and collaborative driving. These advantages have been verified by research programs such as PATH in the United States [6], SARTRE in Europe [7], and Energy-ITS in Japan [8]. The multiple-vehicle cooperative mode has completely changed the operation mode of the transportation system.

Cooperative adaptive cruise control (CACC) is based on the idea of multiple-vehicle cooperation, which extends adaptive cruise control (ACC) with cooperative maneuvers by ICVs [9,10]. In CACC systems, ICVs share their own parameters with other ICVs in the network by V2X communications [11,12,13,14], which is realized in an autonomous manner without management. CACC can shorten the inter-vehicle distance, improve system stability, and make dynamic traffic flow more manageable. The question of how to design a control system with high accuracy, rapid response, and good robustness so that safety, convenience, and energy conservation can be achieved has emerged as an urgent issue for CACC control research.

In the literature, research on CACC control primarily focuses on the problem of longitudinal cooperative driving in straight scenes [15,16,17]. However, in the real traffic scene, the vehicles in the platoon will inevitably encounter lateral movement when all vehicles are arranged in a row to move along the desired trajectory with the desired inter-vehicle distance [18]. The existing CACC control algorithm has less research on the lateral control of the vehicle. Due to the existence of the lateral error of the vehicle, the longitudinal velocity is uncertain along the direction of the expected trajectory. Especially in the lane-changing or turning traffic scenario, if the lateral movement of the vehicle is ignored, the following vehicles will fail to follow, and the downstream vehicles will also experience a chain reaction [19]. In addition, on the premise of fully considering the lateral movement of the vehicle, the coupling effects of longitudinal and lateral vehicle movement can not be ignored [20]. In existing research, by setting up the longitudinal and lateral coupling dynamic model, and then using the complex control algorithm such as optimization to design the controller, the control law is complex to solve and the calculation efficiency is poor. In this paper, we propose an adaptive cooperative cruise control algorithm based on the Frenet frame to deal with the aforementioned issues. The main contributions are as followers:

(1) In order to eliminate the coupling effect of longitudinal and lateral movement of the vehicle, the projection transformation of vehicle movement is carried out. The CACC control problem is decoupled into longitudinal control along the reference trajectory and lateral control away from the reference trajectory. In order to describe the vehicle movement state after projection transformation, the vehicle’s kinematic model is set up based on the Frenet frame, which is convenient for dimension reduction control.

(2) Under the Frenet frame, we design a control algorithm based on the decoupled model of longitudinal and lateral dynamics. An exponential convergence strategy is adopted for the longitudinal control. A sigmoidal function is used to assign the weight according to the state information of the leader and the front vehicle, so that the node vehicle can keep the inter-vehicle distance to the desired value. For lateral control, the nonlinear group dynamics equation is linearized using the chain structure. The lateral control law is obtained by using the linear control idea and Lyapunov’s method. The paper also gives proof of system stability.

(3) A comparison experiment of different velocities in the lane-changing scene and different reference trajectories at the same velocity are performed. The comparison results show that the group can track the reference trajectory quickly and is robust to the changes in curvature and speed during the driving process.

The rest of this paper is organized as follows. Section 2 presents related work. The vehicle dynamics model is designed under the Frenet frame in Section 3. Section 4 introduces how to design control laws for longitudinal and lateral directions. The simulation verification is implemented and the results are analyzed in Section 5. Lastly, conclusions are given in Section 6.

## 2. Related Work

The basis for the Frenet frame is a reference line in the road [21,22]. The movement of the vehicle depicted in the Frenet frame is independent of the absolute position of the vehicle in the curved road scene, but related only to the vehicle’s position with respect to the reference trajectory [23,24]. This will decouple the vehicle’s movement from a complex two-dimensional in Cartesian coordinates to a simple one-dimensional in two directions which are perpendicular and parallel to the reference trajectory. Thus, the transformation from a high-dimensional computational model to a low-dimensional computational model can be performed. It is possible to improve the performance of the decision-making model, as well as improve the efficiency of the control system. As a result, a growing number of researchers are choosing to build the vehicle trajectory planner controller and trajectory tracking controller based on the Frenet frame. Werling et al. designed a parameter adaptive trajectory planning algorithm based on the Frenet framework, which can meet the requirements of speed maintenance, lane maintenance, and obstacle avoidance during vehicle driving [25]. The description of the vehicle movement trajectory based on the Frenet frame was investigated by WANG SJ et al., which makes the trajectory fitting calculation simple, greatly simplifies the movement description model, and improves calculation efficiency [26]. WANG W et al. investigated the autonomous vehicle trajectory tracking problem based on the Frenet coordinate system [27]. The vehicle movement model derived from the Frenet coordinate system can also be used directly in the multiple-vehicle driving problem to represent the position relationship of the node vehicles with respect to the leader [28,29].

In recent years, the research on CACC mainly focuses on longitudinal single-lane control, including stability analysis [30,31,32,33,34], communication problem analysis [35,36,37,38], heterogeneous problem analysis [39,40,41,42], etc. The control algorithm is mainly aimed at improving the original single-vehicle control algorithm, such as PID control, sliding mode control, and distributed model predictive control. For example, based on ACC control, Vanderwerf et al. designed a linear CACC controller. The controller which aims at tracking the acceleration of the front vehicle has considered the influence of the acceleration of the front vehicle [43]. Ploeg J et al. designed a CACC controller based on the PD control, and adopted the frequency response strategy to investigate the effect of different headway on the stability of the platoon [44]. Sawant J et al. proposed a disturbance observer-based sliding mode control for the control of a cooperative adaptive cruise control system and the proposed strategy addressed practical issues such as the unavailability of preceding vehicle acceleration and range of uncertainty in the vehicle dynamics [45]. Fabrizio D R et al. proposed a sliding mode-based estimator ensuring the finite time convergence of the acceleration estimation error, and hence, ensuring CACC functionality in degraded situations [46]. Zhang Y et al. proposed a space-domain CACC control algorithm, which is more robust. The stability of the system can be guaranteed as long as the headway between vehicles is larger than the communication delay [47]. Van et al. proposed a distributed-model predictive control algorithm based on feed-forward control theory, which enables group heterogeneity and improves robustness to packet loss problems in communication process [48]. Lin Y et al. proposed an adaptive neuro-fuzzy predictor-based control (ANFPC) approach with integrated automotive radar and vehicle-to-vehicle communication for the design of the CACC system, which concerns not only the safety and riding comfort of a vehicle but also enhances its fuel efficiency [49].

In contrast, when the real road is a curved road, vehicle control involves not only longitudinal movement but also lateral movement. Less attention has been paid to lateral control of the vehicle in the literature. LI Y et al. proposed a longitudinal and lateral controller for platoon driving based on the theory of coherence. The nonlinear controller is designed based on the position error with respect to the leader [19]. Since the controller cannot accurately represent the true longitudinal and lateral vehicle errors, in the variable curvature trajectory tracking scene, they may not be able to follow the desired trajectory normally. An extended look-ahead control method was proposed by BAYUWINDRA et al. for designing longitudinal and lateral nonlinear controllers [50]. When the front and rear vehicles are relatively far apart, however, the latter may swerve too quickly. On the other hand, when the curvature of the road is large, the coupling characteristics of longitudinal and lateral movement of vehicles are evident. Sun Z et al. proposed a novel coordinated longitudinal and lateral control algorithm for the intelligent vehicle to address the problem of the imbalance of tracking accuracy and vehicle stability during the trajectory tracking process. This makes the controller maintain high trajectory tracking accuracy and good robustness in different driving conditions [51]. Kianfar et al. proposed a combined lateral and longitudinal direction control method in a designated trajectory. The method which can satisfy the requirements of longitudinal stability and lateral tracking of vehicles on the designated lane makes use of time-domain and frequency-domain control theory [52]. Above all, in the course of solving the longitudinal and lateral control problem in platoon driving, the longitudinal and lateral coupling dynamics model is usually set up first. The complex control algorithms such as optimization are adopted to design the control law. The solution is complex and the calculation efficiency is poor. In this paper, we propose a CACC algorithm that not only considers lateral movement control but also easily solves the problem above.

## 3. Model for Vehicle Dynamics

In this paper, we focus on the design of the CACC control algorithm with lateral control and longitudinal directions. We consider that n connected intelligent vehicles form a platoon for cooperative driving along the desired trajectory with the desired inter-vehicle distance, as illustrated in Figure 1.

Each vehicle exchanges information with adjacent vehicles via wireless communication technology. Taking into account the coupling effect of the vehicle’s longitudinal and lateral movement, the vehicle is projected onto the reference trajectory. The Frenet frame is set up with the start point of the reference trajectory as the origin, the s axis along the reference trajectory, and the y axis perpendicular to the reference trajectory. The control problem of CACC will be decoupled into longitudinal control along the reference trajectory and lateral control away from the reference trajectory. The red vehicle represents the actual vehicle node in the platoon, and the white vehicle represents the projected vehicle node. The vehicle with the maximum value of s is designated as the leader. The n − 1 remaining vehicles are called followers.

The vehicle kinematics model is set up based on the Frenet frame to describe the state of the vehicle after the projection transformation, as shown in Figure 2. Cartesian frame $\left\{O,X,Y\right\}$ and Frenet frame $\left\{O_{s},s,y\right\}$ can be found in the figure. The dashed red line represents the tangent direction of the reference trajectory. The orange dashed line shows the normal direction of the reference trajectory. The blue dashed line shows the right-hand direction under the Cartesian coordinate system. The solid blue line shows the vehicle’s driving direction. Select the central position of the rear wheel $O_{i}$ to identify the vehicle. The Cartesian location coordinates of vehicle_i is denoted as $\left(X_{i},Y_{i}\right)$.

Notations: The orientation of vehicle_i with respect to Cartesian frame is denoted as $\theta_{i}$. The steering angle of vehicle_i is denoted as $\delta_{i}$. The longitudinal velocity of vehicle_i is denoted as $v_{i}$. The wheelbase of the vehicle is denoted as L. The reference trajectory is denoted as C. The projection point of vehicle_i which is closet to reference trajectory is denoted as $M_{i}$. The curvilinear abscissa of vehicle_i is denoted as $s_{i}$. The lateral deviation of vehicle_i from C is denoted as $y_{i}$. The orientation of the tangent to the trajectory at the same point in relation to the absolute reference is denoted as $\theta_{ci}$. The angular deviation of vehicle_i from the trajectory C is denoted as ${\tilde{\theta }}_{i}$. The local curvature is denoted as $c_{i}$. $D_{i}$ is the center of the trajectory arc.

Through the analysis of geometric relationships, we can obtain:(1)$$\frac{D_{i}M_{i}}{D_{i}O_{i}}=\frac{M_{i}Q_{i}}{P_{i}O_{i}}$$
where $M_{i}Q_{i}={\dot{s}}_{i}$. Then we can obtain:(2)$$\frac{1/c_{i}}{1/c_{i}-y_{i}}=\frac{{\dot{s}}_{i}}{v_{i}\cdot \cos{\tilde{\theta }}_{i}}$$

Transforming the equation gives:(3)$${\dot{s}}_{i}=\frac{v_{i}\cdot \cos{\tilde{\theta }}_{i}}{1-y_{i}c_{i}}$$

Meanwhile, we know that $s_{i}=r_{ci}\cdot \theta_{ci}$, $r_{ci}=\frac{1}{c_{i}}$, we can obtain: $\theta_{ci}=s_{i}\cdot c_{i}$, ${\dot{\theta }}_{ci}={\dot{s}}_{i}\cdot c_{i}$.

Given by ${\tilde{\theta }}_{i}=\theta_{i}-\theta_{ci}$, and combined with the kinematic model of the vehicle in the Cartesian coordinate system [53], in the Frenet frame, the dynamic vehicle model of the group is obtained as follows:(4)$$\left\{\begin{array}{l} {\dot{s}}_{i}=v_{i}\cdot \chi_{i}\\ {\dot{y}}_{i}=v_{i}\cdot \sin{\tilde{\theta }}_{i}\\ {\dot{\tilde{\theta }}}_{i}=v_{i}(\frac{\tan\delta_{i}}{L}-c_{i}\cdot \chi_{i}) \end{array}\right.$$
where $\chi_{i}=\frac{\cos{\tilde{\theta }}_{i}}{1-y_{i}c_{i}}$.

A more general form of the model is expressed as:(5)$${\dot{\xi }}_{i}=f_{i}(\delta_{i},u_{i})$$
where $\xi_{i}={\left[s_{i},y_{i},{\dot{\tilde{\theta }}}_{i}\right]}^{T}$ is the variable of state, $u_{s,i}=v_{i}$ is the longitudinal control input, $u_{y,i}=\delta_{i}$ is the lateral control input.

## 4. Design of Control Strategy

The cooperative adaptive cruise control (CACC) algorithm proposed in this paper will design control laws for longitudinal and lateral directions respectively.

### 4.1. Longitudinal Control

The objective of longitudinal control is to ensure that the space of vehicles in the longitudinal direction can be stabilized to the desired inter-vehicle distance. We adopt the Predecessor-leader Following (PLF) communication topology [14] as presented in Figure 3 of this paper. The vehicles of the platoon can obtain information from both the front and the leader at the same time.

We will treat the front vehicle information and the leader vehicle information received by the i-th vehicle respectively. The first step is to study the control of the i-th vehicle receiving information from the front vehicle. The leader-follower structure of vehicles in the platoon under Frenet frame is shown in Figure 4. The curvilinear abscissa of vehicle_i is $s_{i}$ and the curvilinear abscissa of vehicle_i−1 is $s_{i-1}$. The inter-vehicle distance is d.

The longitudinal displacement between the i-th vehicle and the front vehicle represents the longitudinal error as follows:(6)$$e_{i}^{i-1}=s_{i-1}^{i-2}-s_{i}^{i-1}-d$$
where $e_{i}^{i-1}$ is the spacing error between the i-th vehicle and the front vehicle in the longitudinal direction under the Frenet frame, d is desired inter-vehicle distance. Here a fixed vehicle spacing strategy is adopted, i.e., d is a constant.

The derivation of $e_{i}^{i-1}$ goes as follows:(7)$${\dot{e}}_{i}^{i-1}={\dot{s}}_{i-1}^{i-2}-{\dot{s}}_{i}^{i-1}=v_{i-1}^{i-2}\cdot \chi_{i-1}-v_{i}^{i-1}\cdot \chi_{i}$$
where $v_{i}^{i-1}$ represents the velocity of the i-th vehicle under the condition that only the information from the front vehicle can be obtained, $v_{i-1}^{i-2}$ is similar.

From Equation (7), we can obtain:(8)$$v_{i}^{i-1}=\frac{1}{\chi_{i}}\left(v_{i-1}^{i-2}\cdot \chi_{i-1}-{\dot{e}}_{i}^{i-1}\right)$$

In order to achieve the control objective $\underset{t\to \infty }{\lim}e_{i}^{i-1}(t)=0$, the exponential convergence method is used:(9)$${\dot{e}}_{i}^{i-1}=-k_{1}\cdot e_{i}^{i-1}$$
where $k_{1}>0$. Then substituting the Equation (9) into Equation (8), we can obtain:(10)$$v_{i}^{i-1}=\frac{1}{\chi_{i}}\left(v_{i-1}^{i-2}\cdot \chi_{i-1}+k_{1}\cdot e_{i}^{i-1}\right)$$

Next, we study the control strategy of the i-th vehicle when it receives the information of the leader. The longitudinal displacement between the i-th vehicle and the leader represents the longitudinal error as follows:(11)$$e_{i}^{1}=s_{1}-s_{i}^{1}-i\cdot d$$
where $e_{i}^{1}$ is the spacing error between the i-th vehicle and the leader in the longitudinal direction under the Frenet frame.

The derivation of $e_{i}^{1}$ goes as follows:(12)$${\dot{e}}_{i}^{1}={\dot{s}}_{1}-{\dot{s}}_{i}^{1}=v_{1}\cdot \chi_{1}-v_{i}^{1}\cdot \chi_{i}$$
where $v_{1}$ is the speed of the leader.

From Equation (12), we can obtain:(13)$$v_{i}^{1}=\frac{1}{\chi_{i}}\left(v_{1}\cdot \chi_{1}-{\dot{e}}_{i}^{1}\right)$$

In order to achieve the control objective $\underset{t\to \infty }{\lim}e_{i}^{1}(t)=0$, the exponential convergence method is used:(14)$${\dot{e}}_{i}^{1}=-k_{2}\cdot e_{i}^{1}$$
where $k_{2}>0$. Then substituting the Equation (14) into Equation (13), we can obtain:(15)$$v_{i}^{1}=\frac{1}{\chi_{i}}\left(v_{1}\cdot \chi_{1}+k_{2}\cdot e_{i}^{1}\right)$$

If we consider that the i-th vehicle obtains information from the front vehicle and the leader at the same time, it is necessary to consider the weight distribution between $v_{i}^{i-1}$ and $v_{i}^{1}$, as shown in Figure 5.

If the current vehicle is close to the front vehicle, it is necessary to take more consideration of the speed and position of the front vehicle in order to prevent a collision. At the same time, when the current vehicle is away from the front vehicle, it is necessary to take more consideration of the speed and position of the leader in order to reach the designated position as quickly as possible. In this paper, we adopt a sigmoidal function for the weight distribution. The longitudinal control law can thus be obtained:(16)$$v_{i}=\omega \cdot v_{i}^{1}+(1-\omega )\cdot v_{i}^{i-1}$$
where $\omega =\frac{1}{1+e^{-a_{i}\cdot e_{i}^{i-1}}}$ is the weighting coefficient, $\alpha_{i}>0$ is the regularization parameter.

### 4.2. Lateral Control

The objective of lateral control is to make the lateral error approaches zero. As the control method of CACC has decoupled lateral and longitudinal under the Frenet frame, it is sufficient to control the steering angle of the front wheel to complete the corresponding control objective.

Set z as an arbitrary variable with the following chain system:(17)$$\left\{\begin{array}{l} {\dot{z}}_{1}=u_{1}\\ {\dot{z}}_{2}=z_{3}\cdot u_{1}\\ {\dot{z}}_{3}=u_{2} \end{array}\right.$$

Consider the following variable substitution $z_{1}=s$, $z_{2}=y$, and we can get:(18)$$\left\{\begin{array}{l} {\dot{z}}_{1}=u_{1}={\dot{s}}_{i}=v_{i}\cdot \chi_{i}\\ {\dot{z}}_{2}=z_{3}\cdot u_{1}={\dot{y}}_{i}=v_{i}\cdot \sin{\tilde{\theta }}_{i}\\ z_{3}=\frac{{\dot{z}}_{2}}{{\dot{z}}_{1}}=(1-y_{i}\cdot c_{i})\cdot \tan{\tilde{\theta }}_{i} \end{array}\right.$$

The derivation of $z_{3}$ goes as follows:(19)$${\dot{z}}_{3}=(-{\dot{y}}_{i}\cdot c_{i}-y_{i}\cdot \frac{dc_{i}}{ds}\cdot \dot{s})\cdot \tan{\tilde{\theta }}_{i}+\frac{\tilde{\theta }}{\chi_{i}\cdot \cos{\tilde{\theta }}_{i}}$$

Above all, Equation (17) can be transformed into:(20)$$\left\{\begin{array}{l} {\dot{z}}_{1}=u_{1}=\dot{s}=v_{i}\cdot \chi_{i}\\ {\dot{z}}_{2}=z_{3}\cdot u_{1}={\dot{y}}_{i}=v_{i}\cdot \sin{\tilde{\theta }}_{i}\\ {\dot{z}}_{3}=u_{2}=(-{\dot{y}}_{i}\cdot c_{i}-y_{i}\cdot \frac{dc_{i}}{ds}\cdot \dot{s})\cdot \tan{\tilde{\theta }}_{i}+\frac{{\tilde{\theta }}_{i}}{\chi_{i}\cdot \cos{\tilde{\theta }}_{i}} \end{array}\right.$$

The chain system described in Equation (17) is similar to the linear system, and the control law is selected as:(21)$${\dot{z}}_{3}=u_{2}=-u_{1}\cdot \gamma_{1}\cdot z_{2}-u_{1}\cdot \gamma_{2}\cdot z_{3}$$

Select Lyapunov function:(22)$$V\left(u\right)=\frac{1}{2}(z_{2}^{2}+\frac{1}{\gamma_{1}}\cdot z_{3}^{2})$$

The derivation of $V\left(u\right)$ goes as follows:(23)$$\begin{array}{l} \dot{V}\left(u\right)={\dot{z}}_{2}\cdot z_{2}+\frac{1}{\gamma_{1}}\cdot {\dot{z}}_{3}\cdot z_{3}\\ =z_{3}\cdot u_{1}\cdot z_{2}+\frac{1}{\gamma_{1}}\cdot (-u_{1}\cdot \gamma_{1}\cdot z_{2}-u_{1}\cdot \gamma_{2}\cdot z_{3})\cdot z_{3}\\ =-\frac{\gamma_{2}}{\gamma_{1}}\cdot u_{1}\cdot z_{3}^{2} \end{array}$$
where $\gamma_{1}>0$, $\gamma_{2}>0$, $u_{1}=\dot{s}\geq 0$.

Thus, when $\dot{V}\left(u\right)\leq 0$, there exists a control law $u_{2}=-u_{1}\cdot \gamma_{1}\cdot z_{2}-u_{1}\cdot \gamma_{2}\cdot z_{3}$ that causes the chain system to satisfy Lyapunov stability. In order to complete the control objective, the node vehicle can adjust the lateral error to approach zero.

Substitute Equation (21) and ${\dot{\tilde{\theta }}}_{i}=v_{i}(\frac{\tan\delta_{i}}{L}-c_{i}\cdot \chi_{i})$ in the 3rd term of Equation (20) to derive the lateral control input of the node vehicle as follows:(24)$$\delta_{i}=arc\tan(L[((\sin{\tilde{\theta }}_{i}\cdot c_{i}+y_{i}\cdot \frac{dc_{i}}{ds}\cdot \chi_{i})\tan{\tilde{\theta }}_{i}-\chi_{i}\cdot \gamma_{1}\cdot y_{i}-\gamma_{2}\cdot \sin{\tilde{\theta }}_{i})\cdot \chi_{i}\cdot \cos\tilde{\theta }+\chi_{i}])$$

## 5. Simulation Analysis

Unlike the traditional CACC control algorithm which includes only longitudinal control, lateral control is also considered in the control algorithm proposed in this paper. For this reason, a common lane changing scene is chosen for verification.

### 5.1. Trajectory Generation and Parameter Calculation

In this paper, the B-spline interpolation fitting method is used for segment fitting of reference trajectory. Assume the degree of B-spline curve is k. The set of control points $p_{n}={\left[\begin{array}{cc} X_{n} & Y_{n} \end{array}\right]}^{T}$ is given by where n=1,⋯,N, and N is the total number of control points. (N+1)−(k−1) continuous spline curve is used to generate the required trajectory. Let $S_{g}(u)$ be an interpolating function:(25)$$S_{g}(u)=U\cdot M\cdot \left[\begin{array}{cc} X_{n} & Y_{n}\\ X_{n+1} & Y_{n+1}\\ \vdots & \vdots \\ X_{n+k-1} & Y_{n+k-1} \end{array}\right]$$
where g=1,2,⋯,(N+1)−(k−1) and $M\in R^{k\times k}$.u is the internal parameter of the interpolation function and $u\in \left[\begin{array}{cc} 0 & 1 \end{array}\right]$, U is a polynomial matrix about u and $U=\left[\begin{array}{ccccc} u^{k-1} & u^{k-2} & \cdots & u & 1 \end{array}\right]$.

For a given order k of B-spline curve, the matrix M is always constant:(26)$$M_{i+1,j+1}=\frac{1}{(k-1)!}C_{k-1}^{i}\sum_{m=j}^{k-1}{\left(k-(m+1)\right)}^{i}{\left(-1\right)}^{m-j}C_{k}^{m-j}$$
(27)$$C_{k-1}^{i}=\frac{\left(k-1\right)!}{i!\left(\left(k-1\right)-i\right)!}$$
(28)$$C_{k}^{m-j}=\frac{k!}{(m-j)!\left(k-\left(m-j\right)\right)!}$$
where i=0,⋯,k−1, j=0,⋯,k−1, m=j,⋯,k−1.

Given k=6, Equation (25) can be converted into the following form:(29)$$\left\{\begin{array}{l} X_{i}(u)=a_{5}u^{5}+a_{4}u^{4}+a_{3}u^{3}+a_{2}u^{2}+a_{1}u+a_{0}\\ Y_{i}(u)=b_{5}u^{5}+b_{4}u^{4}+b_{3}u^{3}+b_{2}u^{2}+b_{1}u+b_{0} \end{array}\right.$$
where $a_{0},a_{1},a_{2},a_{3},a_{4},a_{5},b_{0},b_{1},b_{2},b_{3},b_{4},b_{5}$ are the fitting coefficients, which can be solved by the method in reference [54].

Choose a series of control points, and use the method described above to generate the desired reference trajectory. In order to reflect the integrity of the lane-changing process, 100 m straight-line driving sections are added before lane changing, 50 m straight-line driving sections are added after lane changing. As a result, 1580 control points are selected in the lane change scene. The desired reference trajectory for driving is illustrated in Figure 6.

By matrix transformation of Equation (29), we can obtain:(30)$$r_{i}(u)={\left[\begin{array}{cc} X_{i}(u) & Y_{i}(u) \end{array}\right]}^{T}$$

Mathematically, we can get the curve length $s_{i}(u)$ as:(31)$$s_{i}(u)={\int_{0}^{a}\left\|r_{i}^{\prime }(u)\right\|}_{2}d_{u}.\;0\leq a\leq 1$$

Note that the expression $r_{i}^{\prime }(u)$ is tangent vector associated with curve $r_{i}(u)$. Also, the value of parameter a is known to vary from 0 to 1 for all the constituent curves.

The lateral deviation $y_{i}(u)$ between the node vehicle and $M_{i}$ can be computed by utilizing the following relation:(32)$$y_{i}(u)=\sqrt{{\left(X_{M}-X_{i}\right)}^{2}+{\left(Y_{M}-Y_{i}\right)}^{2}}$$

Note the angular deviation ${\tilde{\theta }}_{i}(u)$ can be computed by using the following relation:(33)$${\tilde{\theta }}_{i}(u)=\theta_{i}-\arctan\frac{{Y^{\prime }}_{i}(u)}{{X^{\prime }}_{i}(u)}$$

Curvature $c_{i}(u)$ and derivation curvature $\frac{dc}{ds}$ can be respectively computed as follows:(34)$$c_{i}(u)=\frac{{X^{\prime }}_{i}(u){Y^{\prime \prime }}_{i}(u)-{Y^{\prime }}_{i}(u){X^{\prime \prime }}_{i}(u)}{{\left({X^{\prime }}_{i}{\left(u\right)}^{2}+{Y^{\prime }}_{i}{\left(u\right)}^{2}\right)}^{3/2}}$$
(35)$$\frac{dc}{ds}=\frac{dc}{du}\cdot \frac{du}{ds}$$

### 5.2. Numerical Simulation

We assume that there are 5 vehicles in the group and that these vehicles are isomorphic to one another. A table of simulation parameters is provided in Table 1. The wheel base is 1.5 m in diameter. The desired inter-vehicle distance is 3.5 m. The leader has an initial velocity of 15 m/s. The follower’s initial speed is generated according to Equation (16).

The group’s actual driving trajectory is shown in Figure 7. It can be seen from the figure that although the group does not initially follow the desired reference trajectory and the inter-vehicle distance is different, all five vehicles are able to follow the trajectory soon and the subsequent trajectory nearly coincides. This demonstrates that the algorithm’s trajectory-tracking effect is good.

The change in lateral control quantity is illustrated in Figure 8. The simulation results shown in Figure 8 demonstrate that in order to quickly track the desired reference trajectory, the steering angle of the vehicle changes greatly initially. This phenomenon occurs because the initial positions of the vehicles in the group are different. When following the straight section of the desired reference trajectory, the steering angle of the vehicles is nearly zero. The lane changing occurs around the 6th second and the change in steering angle during the lane change is shown in the magnified portion of Figure 8.

The longitudinal control quantity change is plotted in Figure 9. The results of the simulation in Figure 9 demonstrate that the leader always maintains a constant initial velocity during cooperative adaptive cruise driving. The follower generates a suitable initial longitudinal velocity according to Equation (16). The respective velocities are 16.8 m/s, 13.8 m/s, 7.5 m/s, and 10.8 m/s, respectively. After about 7 s, all of the group’s vehicles track the leader’s speed. The tracking speed can be regulated by $k_{1}$ and $k_{2}$.

In this paper, we adopt the fixed vehicle spacing control strategy. Figure 10 shows the change in distance between vehicles during cooperative adaptive cruise driving. From the figure, it can be seen that vehicles in the group can achieve the preset inter-vehicle distance at high speed. If the curvature of the road changes, the system can still maintain a fixed inter-vehicle distance which indicates that there is no collision in the entire driving process and that the safety of the group can be guaranteed.

Figure 11 shows the change in lateral deviation during cooperative adaptive cruise driving. From the figure, it can be seen that the initial lateral deflection is 1 m. It quickly approaches zero after a period of adjustment and the absolute value of the lateral deviation is relatively low in the lane-changing process, proving the efficiency of the algorithm in lateral control.

In order to verify the robustness of the algorithm to different velocities and trajectories, we perform two groups of comparative experiments under the same control settings.

The first comparative experiment investigated the performance of the algorithm when the leader’s velocity is 20 m/s under the same lane-changing scenario. Figure 12 is the simulation result of trajectory tracking and Figure 13 is the graph of inter-vehicle distance changes. We use these results to observe the safety and tracking performance during the driving process.

In the second comparative experiment, we investigated cooperative driving under the turning scene. Figure 14 shows the simulation results of tracking different trajectories with the same velocity and Figure 15 shows the change in inter-vehicle distance.

From the comparison experiment, it can be seen that the spacing error of vehicles in the group is nearly zero longitudinally. When the initial leader velocities are different, only the convergence time is different. Performance parameters such as lateral deviation are nearly unchanged. The time interval in which the follower’s velocity converges to the leader’s velocity is denoted as $t_{v}$. The time interval in which the inter-vehicle distance reaches the preset value is denoted as $t_{D}$. Tabulations of $t_{v}$ and $t_{D}$ are given in Table 2.

Meanwhile, it can be found that the lateral deviation of the vehicle will change depending on the different curvature of the road. As the curvature of the road changes, the lateral deviation will increase correspondingly. However, the overall change range is small and can quickly converge to zero. $d_{dev\_\max}$ in Table 2 show the maximum lateral deviation of the platoon after lane changing and turning away from a straight section.

## 6. Conclusions

In this paper, the control problem of CACC is decoupled into the longitudinal control along the reference trajectory and the lateral control away from the reference trajectory under the Frenet frame by projecting the vehicles on the reference trajectory. The longitudinal and lateral control algorithms are designed, respectively, to solve the lateral and longitudinal control problems in the process of cooperative adaptive cruise driving. Aiming at the longitudinal control problem of the vehicle, the tracking performance of the controlled vehicle to its front one and the leader is guaranteed by satisfying the exponential convergence condition, and the tracking weight is balanced by the sigmoidal function. As for the vehicle lateral control problem, the Lyapunov method is used to design the control algorithm and the proof of stability is presented. Through the comparison test of changing lanes at different velocities and tracking different reference trajectories at the same velocity, we prove that the group can quickly follow the given trajectory using the algorithm. It is robust to the change in curvature and velocity during driving.

In the future, the findings of this project may be extended with an adaptive control strategy so as to accomplish platooning for an initially unknown trajectory. The effects of sliding were ignored and tyre-to-road contact was assumed to be a single point. The communication delays were not considered anywhere in the analysis. Additionally, it was assumed that all geometric parameters were known with exact accuracy whereas in real life it might not be the case. All the conditions mentioned above may be studied in the future to check the performance of the proposed algorithm.

## Figures and Tables

**Figure 1 sensors-23-01888-f001:**
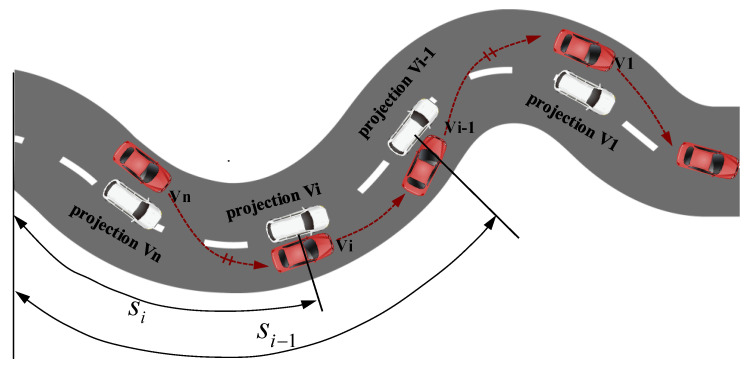
CACC control problem with projection transformation of vehicle node.

**Figure 2 sensors-23-01888-f002:**
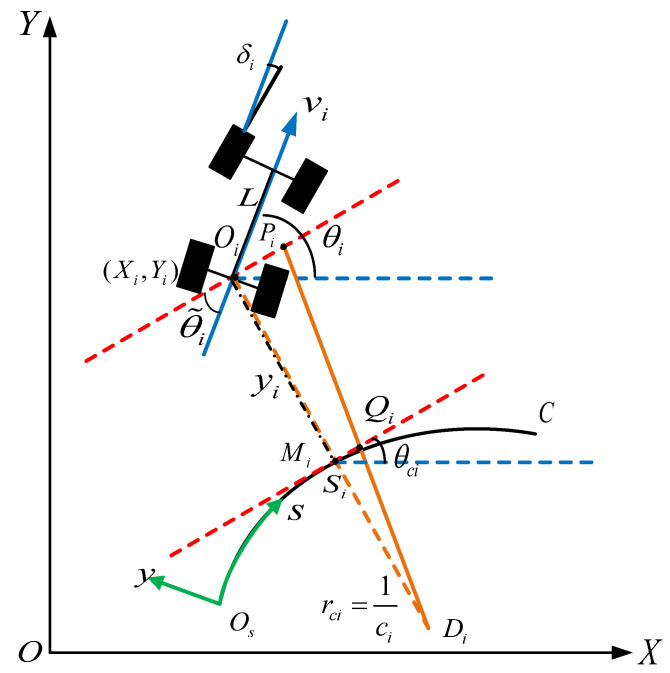
State of vehicle in the Frenet frame.

**Figure 3 sensors-23-01888-f003:**
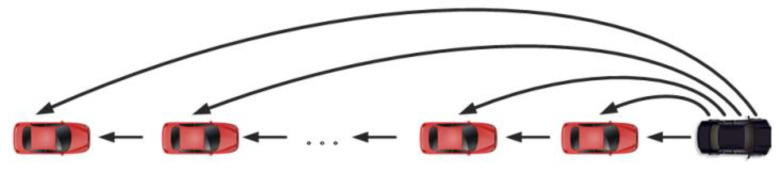
The adopted communication topology of the platoon: PLF.

**Figure 4 sensors-23-01888-f004:**
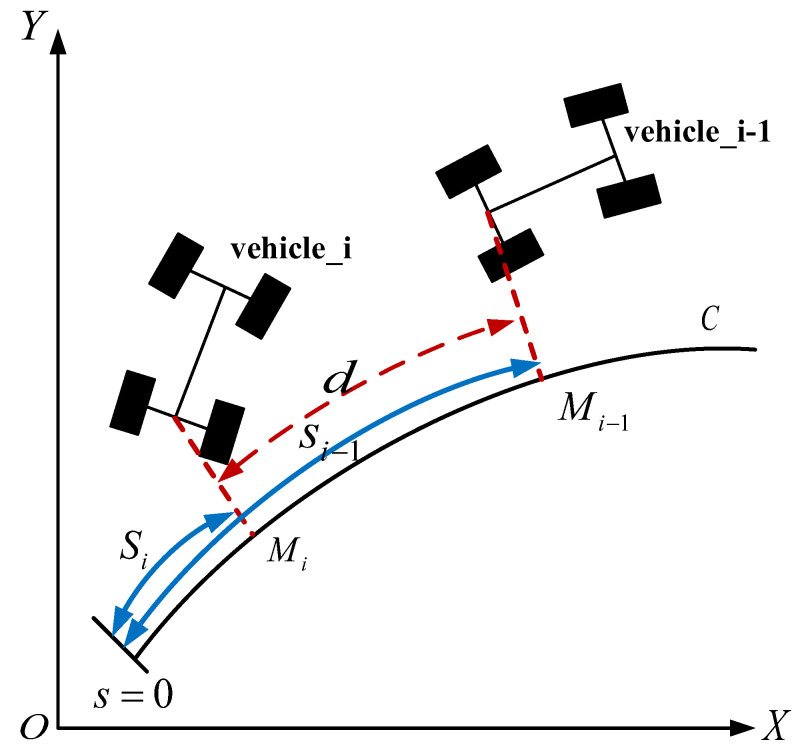
Leader-follower structure of the vehicles in the platoon.

**Figure 5 sensors-23-01888-f005:**
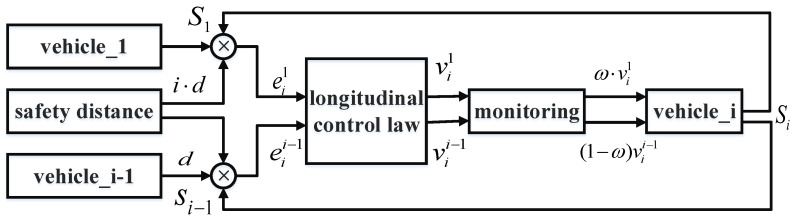
Longitudinal control strategy.

**Figure 6 sensors-23-01888-f006:**
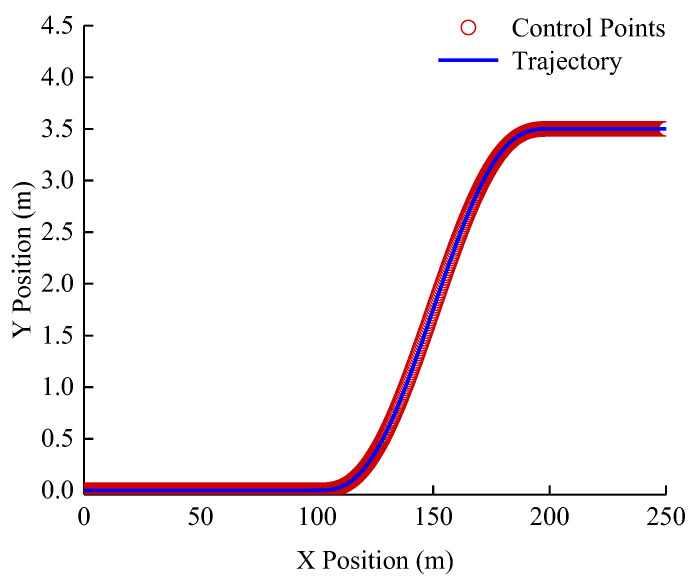
Lane Changing Scene: Desired driving Trajectory of the group.

**Figure 7 sensors-23-01888-f007:**
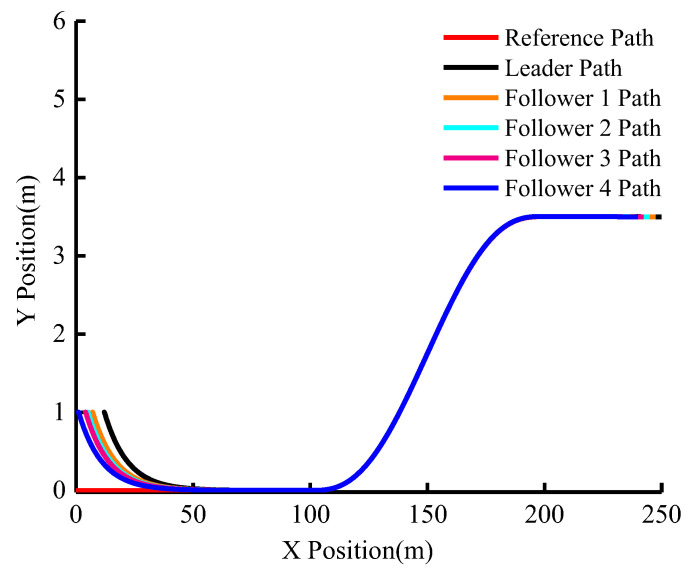
The actual driving trajectory of the group ($v_{leader}=15$ m/s).

**Figure 8 sensors-23-01888-f008:**
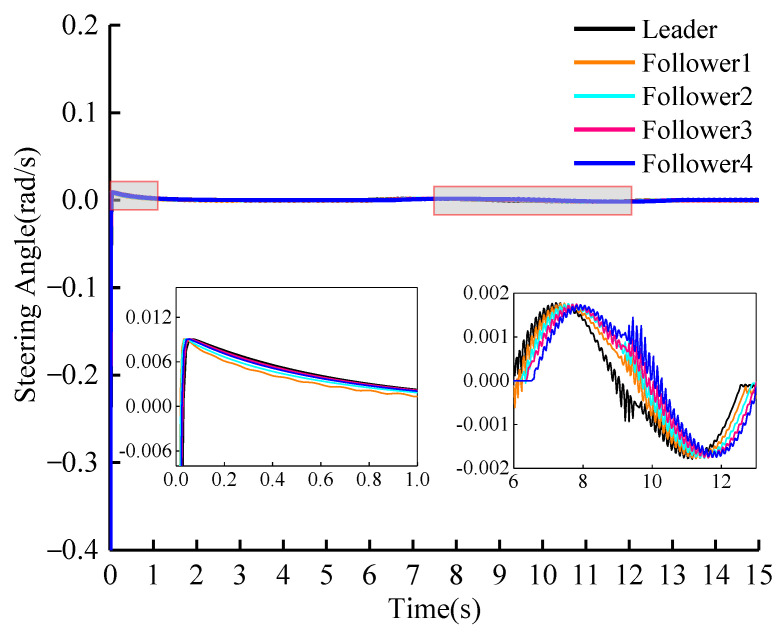
Lateral control of vehicles: steering angle ($v_{leader}=15$ m/s).

**Figure 9 sensors-23-01888-f009:**
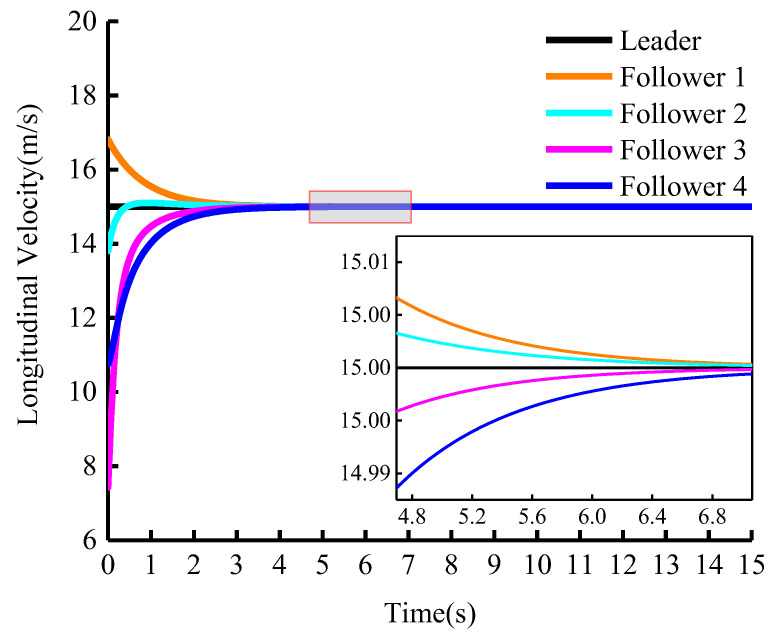
Longitudinal control of vehicles: Longitudinal velocity ($v_{leader}=15$ m/s).

**Figure 10 sensors-23-01888-f010:**
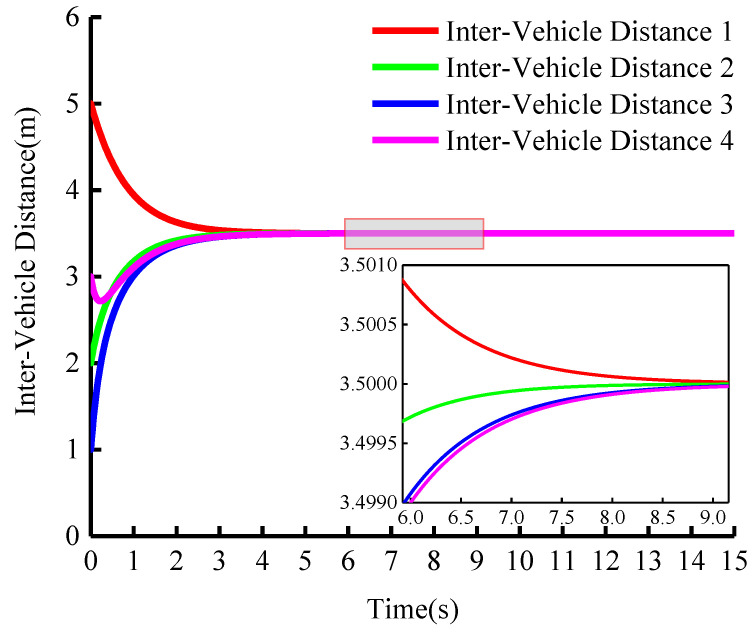
Longitudinal curvilinear distance between vehicles ($v_{leader}=15$ m/s).

**Figure 11 sensors-23-01888-f011:**
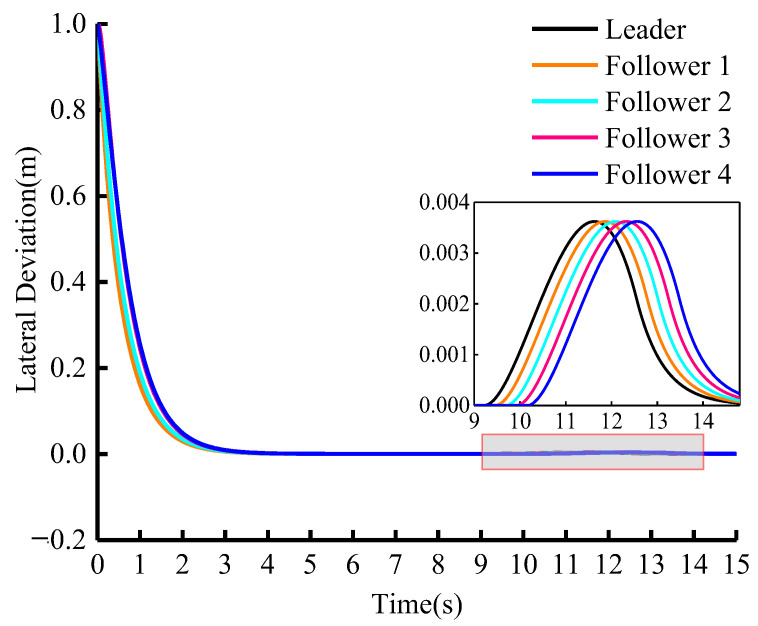
Longitudinal curvilinear distance between vehicles ($v_{leader}=15$ m/s).

**Figure 12 sensors-23-01888-f012:**
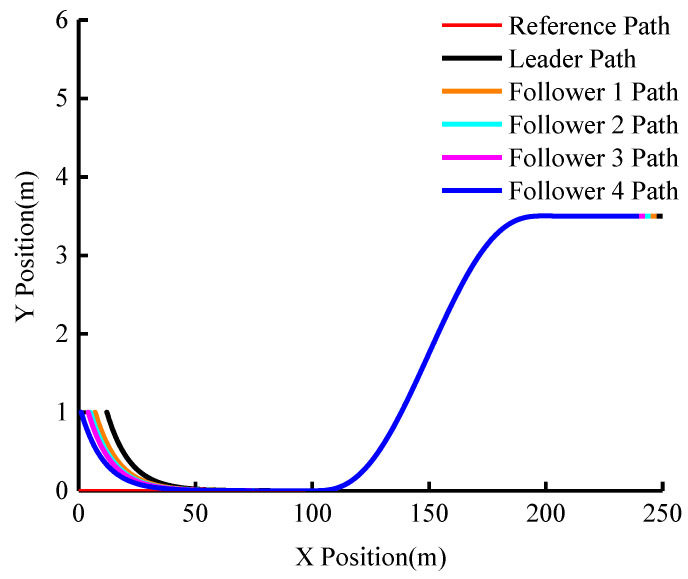
Actual driving Trajectory of the group ($v_{leader}=20$ m/s).

**Figure 13 sensors-23-01888-f013:**
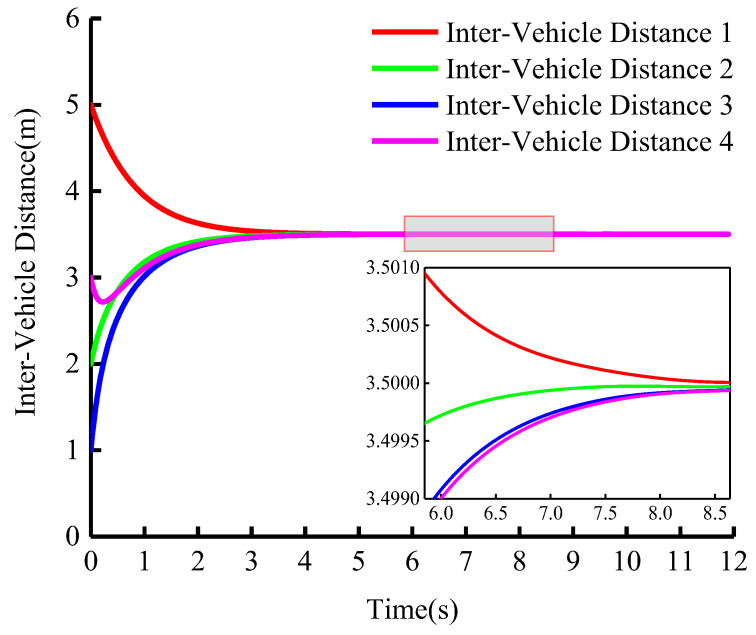
Longitudinal curvilinear distance between vehicles ($v_{leader}=20$ m/s).

**Figure 14 sensors-23-01888-f014:**
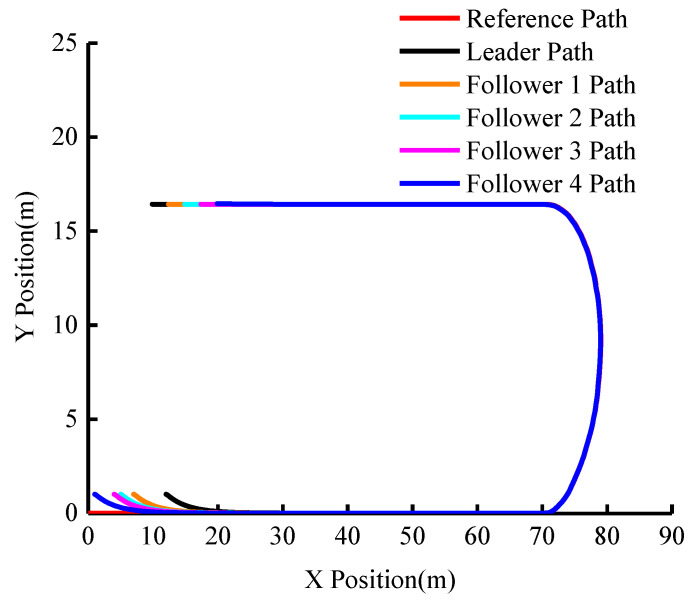
Turning Scene: Actual driving Trajectory of the group ($v_{leader}=15$ m/s).

**Figure 15 sensors-23-01888-f015:**
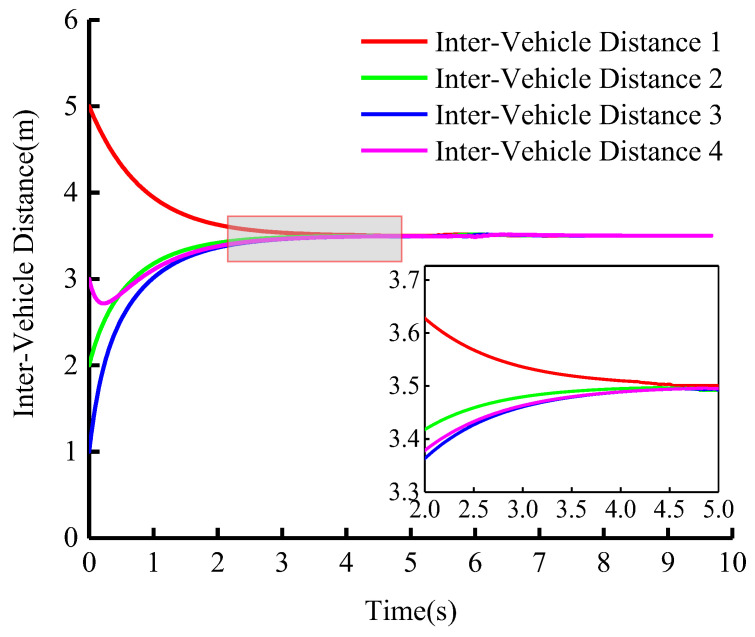
Turning Scene: Longitudinal curvilinear distance between vehicles ($v_{leader}=15$ m/s).

**Table 1 sensors-23-01888-t001:** Parameters in simulation.

Parameter	Value
wheel base (L)	1.5 m
initial position (Leader)	${\left[\begin{array}{cc} 12 & 1 \end{array}\right]}^{T}$ m
initial orientation (Leader)	0 rad
initial position (Follower 1)	${\left[\begin{array}{cc} 7 & 1 \end{array}\right]}^{T}$ m
initial orientation (Follower 1)	0 rad
initial position (Follower 2)	${\left[\begin{array}{cc} 5 & 1 \end{array}\right]}^{T}$ m
initial orientation (Follower 2)	0 rad
initial position (Follower 3)	${\left[\begin{array}{cc} 4 & 1 \end{array}\right]}^{T}$ m
initial orientation (Follower 3)	0 rad
initial position (Follower 4)	${\left[\begin{array}{cc} 1 & 1 \end{array}\right]}^{T}$ m
initial orientation (Follower4)	0 rad
initial velocity (Leader)	15 m/s
longitudinal control parameters ($k_{1}$)	2.8
longitudinal control parameters ($k_{2}$)	1.2
longitudinal control parameters ($\alpha_{i}$)	2
lateral control parameters ($\gamma_{1}$)	8
lateral control parameters ($\gamma_{2}$)	1

**Table 2 sensors-23-01888-t002:** Comparison of performance indexes.

Scene	$v_{0}(m/s)$	$t_{v}(s)$	$t_{D}(s)$	$d_{dev\_\max}(m)$
Lane Changing	15	7.2	7.58	0.0018
	20	6.54	7.72	0.0018
Turning	15	4.02	4.74	0.0815

## Data Availability

Not applicable.

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
