# Peer review of "Research on a Cooperative Adaptive Cruise Control (CACC) Algorithm Based on Frenet Frame with Lateral and Longitudinal Directions"

_sensors, 2023, doi:10.3390/s23041888_

Round 1
Reviewer 1 Report
This is an interesting paper about Cooperative Adaptive Cruise Control. The results are pretty decent. The paper presents an original contribution but some points need improvement. I recommend to be accepted for publication after minor adjustments. In general, I would like the authors to explore the following points:
Feasibility of implementing the algorithm to operate online in real time. I think some attention should be paid to this aspect;
Comment on any limitations of the proposed method. The results show full effectiveness. The authors have not identified any limitations or aspects that could be improved?
I found that the description of the figures regarding the problem description and the method developed is very brief and lacks more details to assist the reader. I found it difficult to follow the control model proposal
The conclusion is not of the same standard as the rest of the paper. Please consider describing better the main findings of the paper and the contributions to the research topic.
Some other aspects that need clarification:
How the designation for the first car (leader) is taken?
On the Fig 1 of [1,2] the control structure considered is presented. I would suggest the authors to include similar approach, presenting the assumed/necessary architecture of the CACC for the proposed method.
Car speeds of 15m/s with a distance of 3.5m, if I understand correctly, seems to be not very realistic.
Section 3 needs to be better presented, I had trouble following the model description. Eq(1) the term D is not presented. The reference trajectory is denoted as C but it is not used.
Fig1 has to be better explained what the colors assigned to the car represent.
Again, the description of Fig2 too brief, please explain in detail. Again the presentation is short and incomplete. For example, what are the blue and red dashed lines? (Figure 4, more details are need)
How do you address in the real case the fact that d is not a constant? Eq7 would have a new term...
Line 217, include reference
Line 270: In real case implementation, how B-spline interpolation fitting method would be implemented for segmental fitting? Theoretically you don't know the complete trajectory, there is no way to interpolate, please comment it.
Could you explain where the parameter values in Tab1 were assigned from?
I would recommend to change the scale of Fig8, maybe keeping only the subpanel.
References:
[1] Wang, Z., Wu, G., & Barth, M. J. (2018, November). A review on cooperative adaptive cruise control (CACC) systems: Architectures, controls, and applications. In 2018 21st International Conference on Intelligent Transportation Systems (ITSC) (pp. 2884-2891). IEEE.
[2] Dey, K. C., Yan, L., Wang, X., Wang, Y., Shen, H., Chowdhury, M., ... & Soundararaj, V. (2015). A review of communication, driver characteristics, and controls aspects of cooperative adaptive cruise control (CACC). IEEE Transactions on Intelligent Transportation Systems, 17(2), 491-509.
Reviewer 2 Report
This paper proposes a cooperative adaptive cruise control (CACC) algorithm based on the Frenet frame. This algorithm decouples the vehicle lateral and longitudinal motion and the authors claim that it improves its efficiency.
The paper is well structured. The quality of the English used makes it challenging however to follow some of the arguments put forward.
The paper is topical and relatively well motivated. The related work discussion, however, includes only very few recent papers. In addition, the claim that the approach improves CACC efficiency is not evaluated at all as the proposed algorithm is not compared with any of its competitors.
Reviewer 3 Report
the paper is well structured, the topic is really topical, the authors did a good overall job, they have to improve the conclusions, they have to integrate them, another aspect to take care of is the introductory part of autonomous vehicles. check the English language from a grammatical point of view. I suggest improving the bibliography:
Managed Lane as Strategy for Traffic Flow and Safety: A Case Study of Catania Ring Road
S Cafiso, A Di Graziano, T Giuffrè, G Pappalardo, A Severino
Sustainability 14 (5), 2915
Reviewer 4 Report
A. Summary:
· The authors propose a control algorithm for cooperative adaptive cruise control that decouples the longitudinal control from the lateral control. To achieve this, the vehicle dynamics in modeled using the frenet frame, and two control laws are developed. For longitudinal control, the aim of the control is to ensure that the follower vehicles track and maintains the inter-vehicle distance between the next vehicle in front as well as the leader vehicle. A control law based on the exponential convergence approach was used. Each vehicle receives velocity information from both leader and the next front vehicle; the effect of the two velocities (front’s and leader’s) on the control of the node vehicle is weighted using a sigmoid function. For the lateral control, the purpose was to reduce the lateral deviation of the node vehicle from the reference road path. The nonlinear dynamic equations were first linearized and a Lyapunov function approach was employed to design a control law with the corresponding input parameter being vehicle’s steering angle.
· In the test simulation, a lane change maneuver and turning case was employed.
The trajectory is generated using a continuous (B-) spline curve was adopted.
· Per the results of the numerical simulation, 4 follower vehicles were able to track the velocity of the lead vehicle and maintain the inter-vehicle distance (3.5m). The lateral distance error was also reduced to zero after a given time during turning motion.
· The labels of most figures are not descriptive enough.
The figures of the simulation results are not clear and detailed enough.
B. Comments on the review
· Equation 31: parameter “q” is not defined.
· Equation 23 could be re-aligned so the complete equation could be seen well.
· Equation 16 : Per understanding the equation should be V^ (i-1) at the top to indicate information from the other adjacent vehicle.
· Figures 1, 3: The label can be more elaborate to communicate what is happening per figure to aid the reader.
· Figure 2: “Kinematic Model of Vehicle in the group” may not be the best description for the diagram.
- The stated reference trajectory “C” in sentence line 173 could be found in the diagram.
· Figure 5 (Longitudinal control)
- The label “weight distribution of longitudinal control” seems may not adequately describe the figure. Weight distribution parameter w is not indicated in the figure.
- Information on what variable or parameters are being returned from “vehicle_i” as feedback to the summation point would be helpful.
- The diagram could be improved to show the weight distribution component, the signals returned from, vehicle_i, and all signal incoming to the summing point leading to the error calculation.
· Sentences / Typo Errors
- The combined sentence from line 43 to line 47 is quite long. This could be restructured to aid understanding.
- Line 274, page 10: The full stop after line 274 could be removed to improve the continuity of the sentence from 273.
Round 2
Reviewer 2 Report
The significant rewrite of the paper, its updated message and the updated state of the art address my previous comments.
A few minor spelling/grammar mistakes remain:
- l 77 followers -> follows
- l l93 are -> is
- l 174 remove "n"